# Oilseed Supplementation Improves Milk Composition and Fatty Acid Profile of Cow Milk: A Meta-Analysis and Meta-Regression

**DOI:** 10.3390/ani12131642

**Published:** 2022-06-26

**Authors:** Genaro Plata-Pérez, Juan C. Angeles-Hernandez, Ernesto Morales-Almaráz, Oscar E. Del Razo-Rodríguez, Felipe López-González, Armando Peláez-Acero, Rafael G. Campos-Montiel, Einar Vargas-Bello-Pérez, Rodolfo Vieyra-Alberto

**Affiliations:** 1Instituto de Ciencias Agropecuarias, Universidad Autónoma del Estado de Hidalgo, Av. Universidad km 1, Tulancingo de Bravo 43600, Mexico; pl429387@uaeh.edu.mx (G.P.-P.); oscare@uaeh.edu.mx (O.E.D.R.-R.); pelaeza@uaeh.edu.mx (A.P.-A.); rcampos@uaeh.edu.mx (R.G.C.-M.); 2Departamento de Nutrición Animal, Facultad de Medicina Veterinaria y Zootecnia, Universidad Autónoma del Estado de México, Instituto Literario 100 Ote, Toluca 50000, Mexico; emoralesa@uaemex.mx; 3Instituto de Ciencias Agropecuarias y Rurales, Universidad Autónoma del Estado de México, Instituto Literario No. 100 Ote, Toluca 50000, Mexico; flopezg@uaemex.mx; 4Department of Animal Sciences, School of Agriculture, Policy and Development, University of Reading, Reading RG6 6EU, UK; e.vargasbelloperez@reading.ac.uk

**Keywords:** CLA, atherogenic index, milk fatty acid, experimental design, roasted, washout

## Abstract

**Simple Summary:**

Milk is the most consumed dairy product in the world and for humans is one of the major sources of beneficial biocomponents. Lipids from oilseeds can be transferred to milk from cows or converted to other biomolecules with nutraceutical effects, resulting in healthier milk. However, there is a great uncertainty with regard to the effect of some variables related to the animal, the seed, the level of inclusion and the characteristics of the diet. The objective of this review was to show the effect of the inclusion of oilseeds in the diet of dairy cows on milk yield, milk components and the fatty acid profile in milk. A systematized search was carried out of published articles with high scientific rigor where the feeding strategy in dairy cows was the inclusion of oilseeds in the diet. Milk from oilseed-fed cows contained a higher amount of unsaturated 18-carbon fatty acids, including omega-3 series, rumenic and vaccenic fatty acids. Overall, supplementation with oilseeds in the cow’s diets increases the concentration of biomolecules in milk with potential positive effects on human health.

**Abstract:**

Oilseed supplementation is a strategy to improve milk production and milk composition in dairy cows; however, the response to this approach is inconsistent. Thus, the aim of this study was to evaluate the effect of oilseed supplementation on milk production and milk composition in dairy cows via a meta-analysis and meta-regression. A comprehensive and structured search was performed using the following electronic databases: Google Scholar, Primo-UAEH and PubMed. The response variables were: milk yield (MY), atherogenic index (AI), Σ omega-3 PUFA, Σ omega-6 PUFA, fat, protein, lactose, linoleic acid (LA), linolenic acid (LNA), oleic acid (OA), vaccenic acid (VA), conjugated linoleic acid (CLA), unsaturated fatty acid (UFA) and saturated fatty acid (SFA) contents. The explanatory variables were breed, lactation stage (first, second, and third), oilseed type (linseed, soybean, rapeseed, cottonseed, and sunflower), way (whole, extruded, ground, and roasted), dietary inclusion level, difference of the LA, LNA, OA, forage and NDF of supplemented and control rations, washout period and experimental design. A meta-analysis was performed with the “meta” package of the statistical program R. A meta-regression analysis was applied to explore the sources of heretogeneity. The inclusion of oilseeds in dairy cow rations had a positive effect on CLA (+0.27 g 100 g^−1^ fatty acids (FA); *p* < 0.0001), VA (+1.03 g 100 g^−1^ FA; *p* < 0.0001), OA (+3.44 g 100 g^−1^ FA; *p* < 0.0001), LNA (+0.28 g 100 g^−1^ FA; *p* < 0.0001) and UFA (+8.32 g 100 g^−1^ FA; *p* < 0.0001), and negative effects on AI (−1.01; *p* < 0.0001), SFA (−6.51; *p* < 0.0001), fat milk (−0.11%; *p* < 0.001) and protein milk (−0.04%; *p* < 0.007). Fat content was affected by animal breed, lactation stage, type and processing of oilseed and dietary NDF and LA contents. CLA, LA, OA and UFA, desirable FA milk components, were affected by type, processing, and the intake of oilseed; additionally, the concentrations of CLA and VA are affected by washout and design. Oilseed supplementation in dairy cow rations has a positive effect on desirable milk components for human consumption. However, animal response to oilseed supplementation depends on explanatory variables related to experimental design, animal characteristics and the type of oilseed.

## 1. Introduction

The increased consumer concern for the diet–health relationship has increased the attention of nutritionists and animal scientists on the health-beneficial effects of individual foods [1]. Bovine milk is considered as one of the most complete foods with a rich variety of essential nutrients and biocomponents [2]. In this sense, the assessment of milk’s fatty acid (FA) profile has been of great importance in recent decades. Milk fat is the richest natural source of conjugated linoleic acid (CLA), with contents that range from 2 to 53.7 mg g^−1^ [3]. The wide variability of CLA content can be explained by several factors such as breed, geographical region, and management; however, the most effective strategy to modulate milk’s FA profile is through dietary manipulation [4,5].

Conjugated linoleic acid is made up of a set of positional and geometric isomers of LA (C18:2 c9c12). CLA is an intermediate product in lipid metabolism and ruminal biohydrogenation, and is naturally found in milk and meat from ruminants [6]. CLA is particularly important because various studies using animal models or cell cultures, and some using humans, have found nutraceutical properties in its consumption, such as antiatherogenic [7], antidiabetogenic [8], anticarcinogenic [9,10,11] effects, and that it allows the modification of body composition [12].

The manipulation of diets fed to dairy cows determines variations in the fat profile of milk; for instance, the inclusion of ingredients with higher contents of monounsaturated fatty acid (MUFA) and polyunsaturated fatty acid (PUFA) increases the CLA [13]. Fat supplementation in dairy cow rations is mostly based on fats, oils and oilseeds from vegetable sources. When agro-climatic and economic conditions allow, oilseed supplementation has been widely used and evaluated due to their lower cost, higher UFA content and protein contribution to the ration compared to other strategies such as the use of oils. However, results from studies that have assessed the effect of oilseed inclusion in dairy cow rations have been inconsistent, which can be associated with differences in the methodological and experimental conditions, such as the inclusion level, content of precursors of basal diets and processing of oilseeds and animal variables which could potentially increase milk yield, fat content and FA transfer from oilseeds to milk [14].

Meta-analysis is a cross-disciplinary statistical framework for estimating the average effect associated with a determined intervention factor from several study outcomes under different experimental conditions; additionally, a random-effect model of meta-analysis can be expanded to a mixed-effect model to assess the sources of heterogeneity through meta-regression [15]. Our objective was to conduct an analytic review to analyze the global effect of supplementing dairy cows with oilseeds on milk yield and chemical composition through a meta-analysis approach. Additionally, the sources of variability between studies that explain the heterogeneity of oilseed supplementation were explored.

## 2. Materials and Methods

### 2.1. Search Strategy

A compressive and structured search of scientific papers focused on evaluating the effect of oilseed supplementation on milk yield and the chemical composition of dairy cows was performed using the following search engines: Google Scholar (http://scholar.google.com), Primo-UAEH (https://www.uaeh.edu.mx/bdigital) and PubMed (https://pubmed.ncbi.nlm.nih.gov). To avoid reviewer bias, the article search was carried out by three field experts. The keywords provided to the field experts were the following: “dairy cow”, “oilseed”, “milk yield”, “CLA”, “grazing”, “indoor”, and “fatty acids”. These were combined by Boolean logic (“AND”, “OR”). Only peer-reviewed articles published in the English language were considered in the current study. Further articles were identified from the list of references of articles found in the first search and were included in the global database.

### 2.2. Eligibility Criteria

The inclusion criteria for selected studies were as follows: (a) studies published in an international peer-reviewed scientific journal (unpublished manuscripts, conference proceedings, and dissertations were hence excluded); (b) studies that use oilseed-supplemented diets and control groups; (c) studies that reported the level of supplementation of oilseed; (d) studies that specified the procedure of randomly assigning animals to each treatment (experimental design); (e) studies that reported the least square means and a measure of data variability (coefficient of variation, standard deviation or standard error) of the control and supplemented groups; and (f) studies that reported the sample size of each group.

### 2.3. Database

A total of 33 studies were selected to be included in the meta-analysis (Figure 1). From the studies that met the inclusion criteria, one reviewer extracted the data for analysis to an Excel spreadsheet (version 2016, Microsoft Corp., Redmond, WA, USA), and the other two reviewers verified them to search for any discrepancies. An Excel spreadsheet was created for each one of the following response variables: milk yield (MY; l day^−1^), fat content (g 100 g^−1^), protein content (g 100 g^−1^), lactose content (g 100 g^−1^), LA (g 100 g^−1^ FA), LNA (g 100 g^−1^ FA), OA (g 100 g^−1^ FA), VA (g 100 g^−1^ FA), CLA (g 100 g^−1^ FA), Σ omega-3 FA (g 100 g^−1^ FA), Σ omega-6 FA (g 100 g^−1^ FA), SFA (g 100 g^−1^ FA), UFA (g 100 g^−1^ FA) and AI (g 100 g^−1^ FA). To explore the sources of between-study variability, the following explanatory variables were a priori selected: breed of cow (Holstein, Jersey and Brown Swiss), lactation stage (first, second and third), type of oilseed (linseed, soybean, rapeseed, cottonseed, and sunflower), processing of oilseed (whole, extruded, roasted, and ground), inclusion level of oilseed, the period (in days) of washout, type of experimental design (longitudinal or crossover), the difference between the basal and supplement diets of NDF, forage rate, LA, LNA and OA.

### 2.4. Meta-Analytical Procedure

A meta-analysis was conducted using the “meta” package version 4.13-0 [16] in R environment for statistical computing R (version 4.0.2; R Core Team; Vienna, Austria). A random-effects model was implemented in the current work, which assumed that the observed difference among studies is a combination of chance and genuine variation in the intervention effects, where each study is a comparison between an experimental (E) group and a control (C) group. Random-effects models were used using the approach proposed by DerSimonian and Laird [17]:(1)$$Y_{ij}^{E}=N\left(\mu_{i}^{E},\;\delta_{i}^{E}\right),\;j=1,\ldots ,\;n_{i}^{E},\;i=1,\ldots ,k,$$
where $Y_{ij}^{E}$ is the *j*th observation in the *i*th experiment from the experimental group assuming that this is normally distributed with a common mean $\mu_{i}^{E}$ and a common variance $\delta_{i}^{E}$. The measure to summarize the observed intervention effect used in the current work was the standardized mean difference (SMD). The SMD was calculated according to the method proposed by Hedges [18], and the pooling of analyzed studies was carried out using the inverse variance weighting. Additionally, 95% confidence intervals (95% CI) were calculated for individual studies and effect size was determined, assuming a standard normal distribution.

The raw mean difference (RMD) was calculated for response variables sharing the same scales, which permits the interpretation of the summary effect under the original measures units [19]. Heterogeneity was assessed through the estimation of between-study random-effects variance (*t*^2^) and the percentage of variability explained by heterogeneity rather than simple variance (*I*^2^ index) [20]. Values of >35% for the *I*^2^ index of random-effects models were deemed as indicative of between-study heterogeneity. The random-effects model does not explain heterogeneity, and can be explored and reduced through subgroup studies and meta-regression [21].

In the current study, to deal with heterogeneity, a meta-regression analysis was performed on all response variables to find the best model that explains the between-study variability of effect size estimations. In the meta-regression, the variable *x* represents the characteristics of the studies (breed, lactation stage, type of oilseed and method of supplementation, inclusion level, washout period and experimental design) that predict the study effect size (${\hat{\theta }}_{\hat{k}}$), as shown in the following model:(2)$${\hat{\theta }}_{\hat{k}}=\theta +\beta x_{k}+\epsilon_{k}+\zeta_{k\;}$$
where ${\hat{\theta }}_{\hat{k}}$ is the observed effect size, *θ* is the intercept, $\beta x_{k}$ is a predictor (or covariate) $x_{k}$ with a regression coefficient *β* (fixed effect), $\epsilon_{k}$ is the sampling error and $\zeta_{k\;}$ is the between-study error (random effect).

The first step of meta-regression analysis is to incorporate all explanatory variables in a full model; secondly, multi-predictor models were manually reduced through a backward selection of variables until all predictors were significant (*p* < 0.05). Mixed-effects regression models (meta-regression analysis) were constructed in the “metafor” package [22]. Models were compared and selected by means of information-theoretic criteria using Akaike’s Information Criterion and Bayesian Information Criterion; additionally, the goodness of fit was evaluated by the coefficient of correlation analogue (*R*^2^) for meta-regression, which was calculated as follows:(3)$$R^{2}=\frac{\tau^{2}{}_{REM}-\tau^{2}{}_{MEM\;}}{\tau^{2}{}_{REM\;}}$$
where $\tau^{2}{}_{REM}$ is the estimated total heterogeneity based on the random-effects model and $\tau^{2}{}_{MEM\;}$ is the total heterogeneity of the mixed-effects regression model. Publication bias was assessed using a graphical approach through funnel plots [23].

## 3. Results

In the current analytic review, the search strategy resulted in 581 articles. Some studies were excluded for several reasons: not being scientific articles or being duplicates, non-English language, published before 1999, combination of various oilseeds, not mainly testing oilseed supplementation, not reporting cows’ ration information, not reporting specific isomers of FA, not including a control group in the experimental design, not reporting a measure of variability of the results and being meta-analyses or reviews. After applying the inclusion and exclusion criteria, 78 trials over 33 articles published between 1999 and 2020 were included in the meta-analysis (Figure 1). The characteristics of studies included in the meta-analysis are depicted in Table 1. In accordance with our results, the most used oilseed in cow rations was linseed (57%), followed by soybean and rapeseed (17 and 13%). The analyzed studies showed a wide range of oilseed supplementation (2.08–17.7%) and most studies used Holstein cows (92%).

### 3.1. Milk Yield and Composition

The mean values of milk yield and milk composition for the control groups and the respective effect sizes of the oilseed-supplemented groups are shown in Table 2. The meta-analysis showed no significant effect of oilseed supplementation on milk yield (SMD = −0.06; *p* = 0.33) and lactose (SMD = 0.07; *p* = 0.27). Oilseed supplementation reduces fat content (SMD = −0.21; *p* = 0.002) with a raw mean difference (RMD) of 0.11 g 100 g^−1^ between the control and supplemented groups. Figure 2 shows a forest plot of the effect of oilseed supplementation on fat content compared with the control (*I*^2^ = 0.53%; *p* < 0.01), grouped according to the type of oilseed supplemented. The supplementation of oilseeds in cows was associated with a reduced protein content, by 0.04 g 100 g^−1^ (*p* = 0.007), compared with a control group (SMD = −0.20; *p* = 0.003). There was evidence of significant heterogeneity among studies for all milk yield and milk composition response variables (*I*^2^ > 40.0).

### 3.2. Fatty Acid Profile

Oilseed supplementation results in an increase in OA and LNA, compared to the control group, with RMDs of 3.44 and 0.28 g 100 g^−1^ FA (*p* = 0.0001) (Table 2). The forest plot shown in Figure 3 reveals a significant effect of oilseed supplementation on VA content, with an increase of 1.03 g 100 g^−1^ FA; additionally, a high heterogeneity (*I*^2^ = 99.7%) was observed for this outcome, which, in accordance with sub-grouping analysis shown in the forest plot, can be primarily attributed to the type of oilseed supplemented. CLA content was increased by 0.27 g 100 g^−1^ FA (*p* = 0.0001) by oilseed supplementation, an enhancement of 50.0% in comparison with control group (Table 2). The between-study response showed high variability in the CLA outcome (*I*^2^ = 99.4%), which can be partially explained by the type of supplemented oilseed, as shown in Figure 4. LA was affected little by oilseed supplementation, with an RMD of 0.09 g 100 g^−1^ FA (*p* = 0.007) and a non-significant SDM (0.16; *p* = 0.25), as well as non-significant values of milk yield and milk composition for the control groups and their respective effect sizes.

The omega-3 PUFA were increased by oilseed inclusion in dairy cow diets, with an RMD of 0.46 (*p* = 0.0001), in comparison with non-supplemented diets. According to the sub-grouping analysis of omega-3 PUFA, the effect size and level of heterogeneity depended on the method of processing oilseed, ranked: extruded > whole > ground > roasted (Figure 5). Otherwise, oilseed diets were not significantly different from the control group with regard to omega-6 PUFA (RMD = −0.01; *p* = 0.78). The pooled estimated effect size showed that feeding with oilseed decreased the SFA content (SMD = −1.73; *p* = 0.0001) by 26.5% in comparison with the control group (Table 2). Supplementing diets with oilseed significantly increased the content of UFA in milk (SMD = 3.79; *p* = 0.0001). On the contrary, AI decreased with oilseed feeding (SMD = −2.47; *p* = 0.0001). For all FA profile outcomes, a high level of heterogeneity was observed among studies (*I*^2^ > 80%).

### 3.3. Meta-Regression

A meta-regression analysis was performed to explain the significant heterogeneity (*I*^2^ > 40%). Eleven explicative variables were studied to explore between-study variability: (1) cow breed, (2) lactation stage, (3) type of oilseed, (4) method of processing oilseed, (5) level of inclusion, (6) duration of washout period to crossover studies, (7) type of experimental design, (8) the NDF difference between supplemented and control groups, (9) the forage difference between supplemented and control groups, (10) the LA difference between supplemented and control groups and (11) the LNA difference between supplemented and control groups.

Increases in fat and omega-3 PUFA contents in the treatment group were associated positively with the Jersey breed. The effect of lactation stage showed a significant relationship with several outcomes, but there was no global trend and the association depends on the output analyzed. Therefore, within the highlighted results, we found that the supplementation of oilseed in cows during the first stage of lactation promotes an increase in protein (β = 1.57; *p* < 0.01) and linoleic FA (β = 0.49; *p* < 0.05) contents, while decreasing LNA (β = −2.4; *p* < 0.05), UFA (β = −7.31; *p* < 0.001) and AI (β = −0.89; *p* < 0.05).

Concerning the type of supplemented oilseed, the inclusion of soybean increases OA (β = 3.54; *p* < 0.05), CLA (β = 1.65; *p* < 0.05) and Σ omega-6 PUFA (β = 0.67; *p* < 0.01). Feeding with sunflower reduced the fat content (β = −0.46; *p* < 0.05) but increased OA (β = 4.92; *p* < 0.01) and CLA (β = 1.96; *p* < 0.05). Cows supplemented with rapeseed were significantly associated with increased values of OA (β = 6.71; *p* < 0.05), Σ omega-3 PUFA (β = 1.35; *p* < 0.05) and UFA (β = 12.25; *p* < 0.05).

The processing of oilseed showed the most consistent influence in almost all response outcomes; only protein and lactose contents were not significantly associated with this explanative variable. A positive relationship was found between ground oilseed supplementation and LA (β = 0.43; *p* < 0.05) and OA (β = 2.63; *p* < 0.05) contents. Whole oilseed supplementation was associated with increases in fat (β = 0.21; *p* < 0.05), OA (β = 2.61; *p* < 0.05), SFA (β = 4.25; *P* < 0.001) and AI (β = 1.23; *P* < 0.001), as well as decreases in milk yield (β = −0.28; *p* < 0.05), CLA (β = −0.18; *p* < 0.05) and UFA (β = −10.98; *p* < 0.05). Finally, the responses of LNA (β = 4.16; *p* < 0.005), OA (β = 4.41; *p* < 0.05), Σ omega-6 PUFA (β = 0.49; *p* < 0.05) and AI (β = 1.81; *p* < 0.001) were higher with roasted-based diets (Table 3).

The contents of OA (β = 0.33; *p* < 0.05), Σ omega-3 PUFA (β = 0.04; *p* < 0.05) and UFA (β = 0.79; *p* < 0.05) in milk increased linearly when increasing supplemental oilseed. A negative dose-dependent response was observed to milk yield (β = −0.05; *p* < 0.05), LA (β = −0.02; *p* < 0.05), LNA (β = −0.12; *p* < 0.05), Σ omega-6 PUFA (β = −0.01; *p* < 0.05), SFA (β = −0.37; *p* < 0.001) and AI (β = −0.12; *p* < 0.05). However, the direction and level of the relationship between intake and effect size differed among the different types of oilseeds supplemented. For instance, as the bubble plot in Figure 6 shows, the global response to oilseed supplementation showed a positive linear relationship between the inclusion level and SMD of UFA outcome; nevertheless, similar effect size values were observed for linseed and rapeseed, with different levels of supplementation. Additionally, a variability of response was observed, associated with the supplementation level within the oilseed group, as shown in Figure 7, where different levels of SMD of the AI outcome were observed with the same levels of supplementation of linseed, which highlights a possible interaction with other explicative variables such as processing and diet characteristics.

Regarding diet characteristics, replacing forage with oilseed has a significant effect on some FA profile outcomes. The results of the meta-regression revealed that an increase in the difference of forage inclusion (supplemented minus control values) reduces the CLA content (β = −0.09; *p* < 0.05) and Σ omega-6 (β = −0.11; *p* < 0.05); additionally, we observed that UFA milk content increased with an increase in NDF difference (β = 0.19; *p* < 0.001). The higher values of LA in the supplemented group were significantly related to the lower values of lactose (β = 0.005; *p* < 0.001), LNA (β = −0.13; *p* < 0.001), OA (β = −0.23; *p* < 0.001) and VA milk (β = −0.04; *p* < 0.05). A negative relationship was observed between the difference of LNA and OA effect size (β = −0.15; *p* < 0.001).

The meta-regression outputs in the experimental design revealed that those longitudinal designs were related to higher values of the effect size for CLA (β = 1.45; *p* < 0.001) and SFA (β = 4.85; *p* < 0.001). Most mixed models explained more than 50% heterogeneity (*R*^2^) for most of the outcome’s variables. Only LA, LNA and CLA had values of *R*^2^ that were less than 33.5% and higher values of *I*^2^ in mixed models (*I*^2^ > 70%). These findings indicate that unknown or not-reported factors from revised studies could affect the response to oilseed supplementation.

## 4. Discussion

The main factor in the exclusion of the meta-analysis was the absence of a control group, resulting in 33 articles with 78 trials. For this study, the Holstein breed was the most used (92%), probably because it is considered the model breed for milk production due to the animals’ characteristics [57]. In total, 51, 38, and 12% of trials were conducted in the first, second and last (third) stages of lactation, respectively, which is congruent with the crossover of experimental design (59%). Flaxseed predominated among the trials (57%), followed by soybean (17%) and rapeseed (13%); similarly, the process mostly used in oilseeds was extrusion followed by the use of whole oilseeds (40 and 35%). The supplementation range was from 20.8 to 177.0 g kg^−1^ DM, which depended mainly on the availability of the ingredient used in each study.

### 4.1. Milk Yield and Composition

The inclusion of oilseeds in the diet did not affect milk yield due to the isoenergetic balance between diets shown in the great majority of trials, thus not compromising the productive behavior of ruminants [58]. Similarly, Rabiee et al. [59] did not show a statistical difference with the use of oilseeds; however, it can increase milk production with the use of a different source of lipids, where the variation is attributed to the dry matter intake and energy content of the diet.

The decrease in milk fat content is affected by the reduction in the action of bacterial fibrolytic enzymes, precursors of acetic fermentation [60], and the production of beta-hydroxybutyrate and some lipogenic enzymes necessary for de novo synthesis in the mammary gland of the main components in milk fat (short- and medium-chain FA) [59,61,62]. Additionally, at the rumen level, UFA has shown toxic effects on some microorganisms [56] by reducing microbial populations and their fermentative activity [63].

Sunflower, cottonseed and soybeans largely led to a reduction in milk fat, which seems to be related to the amount of LA (greater than 50 g 100 g^−1^ FA), although this is not clear. However, LA is thought to be a substrate in the production of trans isomers, which have been related to milk fat depression syndrome [64]. The supplementation of flaxseed and rapeseed had a lower effect on LA content, which can be associated with the saturation by alternative pathways of their major FA [63], resulting in a lower number of intermediate isomers that inhibit the de novo synthesis of FA in milk fat [64].

In cows, it has been estimated that for every 100 g of fat intake, 0.03% of milk protein is reduced [65]. In this review, we assumed that the substitution of energy ingredients in starch-deficient ruminant diets, such as lipids, affects microbial protein synthesis [60], resulting in a decreased supply and absorption of amino acids in the duodenum, which are required for the formation of milk protein [66]. However, it is possible that milk protein is reduced by the relative decrease in ruminal microorganisms [62,63], caused by the presence of UFA in the rumen [56], which are the main components of the FA structure of the evaluated oilseeds.

The obtained heterogeneity for response variables was reduced with the mixed model (meta-regression) in comparison with the random model. Therefore, our results reveal that the used moderating variables covered a considerable proportion of the between-study variation and must be considered in the design of future studies.

### 4.2. Milk Fatty Acid Profile

The increase in OA in milk could be due to the contribution of OA, LA and LNA in the diet through oilseeds, where the final step in the rumen is the saturation to stearic FA, but in the mammary gland it is desaturated by the enzyme delta-9 desaturase to OA [59]; it should be remembered that the ruminal biohydrogenation pathway of LNA does not include the direct synthesis of OA [67]. In the same sense, LNA that managed to escape biohydrogenation in the rumen was reflected by the increase in its content in milk [64]. Similarly, an increase in these FA in milk was observed via the supplementation of soybean, rapeseed, sunflower and flaxseed in the diet of cows [68]. In contrast, Akraim et al. [24] achieved a high intake of LNA in dairy cows through linseed, but this FA was not expressed in milk.

Vaccenic FA is a biohydrogenation intermediate that is related to the incomplete saturation of LA and LNA [64], the major FA of the oilseeds shown in this review. Theoretically, the presence of LA and LNA in the rumen favors the acceleration of the first step in biohydrogenation and reduces the final saturation step towards stearic FA [69], resulting in an increased flow of the VA into the posterior digestive tract and then into the mammary gland to be secreted into milk [64]. VA is a substrate in de novo synthesis in the mammary gland by delta-9 desaturase enzyme activity to produce CLA [70]. Therefore, the increase in VA is more closely related to greater contents of CLA in milk; this fact could explain the higher levels of CLA in animals supplemented with oilseeds found in the current study. However, the level of response depends on the type of oilseed provided, since the greatest effect size was observed for sunflower seed, soybean and flaxseed, which can be associated with the higher contents of LA and LNA (>57%) in these oilseeds. Additionally, intake of VA increases serum CLA levels in humans [71]; both FA (VA and CLA) have been linked to human health with decreased chronic diseases [72], and the intake of CLA and VA has been shown to lead to a reduction in coronary heart disease, low incidence of atherosclerosis [73], decreased risk of hypertension [74], low risk of type II diabetes mellitus [75] and decreased obesity [76]; in addition, CLA has been attributed to reducing cancer induced in laboratory animals [10], and has great potential to improve menopausal symptoms, bone health, sarcopenia and sarcopenic obesity [77].

Including oilseeds in the diet of cows reduces the content of SFA in milk and increases the concentration of UFA, possibly by decreasing the production of volatile fatty acid (VFA) in the rumen, specifically acetic acid, the main substrate for the de novo synthesis of short- and medium-chain SFA [78]. Propionic acid production can be affected by the substitution of starchy energy ingredients and/or non-structural carbohydrates in the diet [79]. The use of oilseeds in ruminant diets can increase propionic acid production and decrease acetic acid production proportionally. This is because of the release of glycerol from triglycerides in lipolysis [80]. This alteration in the rumen ratio of acetic to propionic acids does not affect milk yield but creates the conditions (mainly rumen pH and/or microbial population) for increased UFA content [81]. Long-chain PUFA that escape from biohydrogenation are absorbed by the intestine and increase the ratio of UFA in milk at the expense of a decrease in SFA [58], which may be conditioned by an inhibitory effect of acetyl-CoA carboxylase and fatty acid synthetase enzymes at the mammary gland level [82].

The importance of increasing the proportion of PUFA in milk resides in the fact that human consumption of PUFA can reduce obesity and mortality from cardiovascular problems by up to 30% and decreases the incidence of diabetes by up to 50% [83]; the omega-3 PUFA showed a 20% reduction in mortality in patients with cardiac problems [84]. Our review showed a 62% increase in omega-3 PUFA in cow’s milk with the inclusion of oilseeds as a dietary supplement. In this regard, extruded and ground whole oilseeds showed the greatest effect, possibly due to the greater availability of these PUFA that have escaped ruminal biohydrogenation by being absorbed by the small intestine and secreted in milk; although, the pericarp of these oilseeds plays an important role in the protection of UFA by limiting biohydrogenation [63].

The AI proposed by Ulbricht and Southgate [85] is the sum of hypercholesterolemic SFA content divided by the sum of protective UFA. A low AI reflects milk with a low SFA content [86]; therefore, the consumer reduces the risk of fat contributing to the development of atheroma [87]. In this sense, this review demonstrates that there is an adequate relation between the biocomponents of milk produced by cows supplemented with oilseeds, i.e., FA profile, and so can be considered as a functional food with benefits on the human health. However, the milk composition and the FA profile showed a high variability between studies in response to oilseed supplementation. Therefore, considering the role of covariates through meta-regression analysis is a fundamental step to provide a full understanding of the oilseed supplementation strategy in dairy cows.

### 4.3. Meta-Regression

Our results showed a higher milk fat content in Jersey cows; however, these findings must be cautiously considered due to the reduced number of studies available in this breed. The higher milk fat content found in Jersey cows compared to that of Holstein could be associated with the higher concentration of SFA (lauric, myristic, palmitic, and stearic) and UFA (LA) in Jersey compared to Holstein cows [88]. Carroll et al. [89] suggest that the differences in fat content between cow breeds is associated with the expression and activity of acetyl-CoA carboxylase and mammary stearoyl CoA desaturase, which are responsible of FA and UFA synthesis, respectively. Additionally, several authors reported a positive relationship between the diameter of the milk fat globule and fat content [90,91]. In this sense, the size of the milk fat globule can be explained partly by the greater fat content in Jersey cow´s milk, supported by the fact that fat globules more than 50 mm in diameter are more numerous in Jersey than Holstein cows by a factor of 50 [89].

The composition of cow’s milk changes with the lactation stage [92]. The meta-regression showed an effect of the lactation stage on the response level to oilseed supplementation of some milk components. The response in terms of milk protein to the intervention factor was higher at the beginning and the end of lactation, possibly due to the amount of milk produced [93]. On the other hand, the effect size of oilseed supplementation with regard to fat content was larger when the experimental period was performed in the second stage of lactation, which can be explained by the stronger relationship between energy balance and milk fat synthesis. During the second stage of lactation, after the peak of lactation, there is an increase in dry matter intake, leading to an increased energy supply used for de novo milk fat synthesis [94]. With regard to the negative relationship observed in UFA (LNA and oleic acid) from experiments carried out in the third stage of lactation, our results are in agreement with Stoop et al. [92], who reported a lower content of C18 FA and high proportion of short- and medium-chain FA (C6:0 to C14:0) in milk towards the third stage of lactation. This fact could explain the negative effect of AI in this lactation stage.

The high LA content of sunflower seed (60.4% of FA) affected milk fat content. Supplementing with PUFA-rich sources reduces acetic and butyric fermentation in the rumen and consequently decreases de novo synthesis in the mammary gland and inhibits lipogenic enzymes [95]. Contrary to this, the addition of sunflower to cow diets has a positive effect on the OA content, possibly due to the FA that escape ruminal biohydrogenation and are absorbed by the small intestine to be secreted in milk and/or by the action of the enzyme delta-9 desaturase that has its action in the mammary gland [59]. Additionally, the inclusion of rapeseed and soybean in dairy cow rations results in a positive relationship with the OA content, which can be associated with the fact that these oilseeds have a high concentration of this FA (54.6 g and 22.9 g 100 g^−1^ FA). Additionally, Sterk et al. [68] found a positive relationship with regard to LA content between diet and milk. Thus, the type of oilseed used in cow diets provide the substrates for the different contents of FA in the milk produced and therefore also influences the level of response to supplementation.

The inclusion of rich sources of PUFA in the diet of ruminants has shown increases in the concentration of omega-3 PUFA in meat and milk [14,96]. The type and source of PUFA consumed by the animal can have different impacts on microbial populations and rumen fermentation [97]. These adverse effects of PUFA may be amplified through the use of vegetable oils in the diet compared to oilseeds [98]. Studies using sunflower oil [99] and soybean oil [100] in the diet of dairy cattle reflected an increased concentration of OA in milk, as well as a reduction in short- and medium-chain SFA (C10:0-C16:0). The decrease in VA in milk fat with dietary supplementation of sources of OA, LA and LNA (majority FA in rapeseed, soybean and linseed) could be explained by the decreased synthesis of VA in the ruminal biohydrogenation of PUFA; the level of supplementation was not sufficient to show an effect on this FA and there was a complete biohydrogenation of OA, LA and LNA to stearic FA. The interactions between the type of oilseed and level of inclusion are shown in Figure 6 and Figure 7, which show that the highest levels of response to oilseed supplementation were reached with inclusions of 15% of DM of linseed and 6–7% of DM of rapeseed.

The grinding process exposes the oilseeds’ FA to biohydrogenation in the rumen, being the most affected the UFA and omega-3 PUFA [63], which is in accordance with the results of the current study; however, some FA such as LA can escape the biohydrogenation process, and hence reach the mammary gland [64]. Additionally, as explained above, the OA may escape this hydrogen saturation in the rumen and/or be desaturated from stearic FA in the mammary gland. The unprocessed oilseeds have a negative effect on dry matter intake, thus promoting a decrease in milk yield [57] with increases (β = 0.21; *p* < 0.05) in milk fat content. Unprocessed oilseeds have a negative effect on UFA and positive one on SFA, which may be due to a slower passage rate which allows the rumen microorganisms to saturate PUFA and MUFA of oilseeds [60].

The pericarp of whole oilseeds reaching the small intestine does not allow the absorption of the contained FA [63]. In this sense, the positive effect of SFA and negative effect of UFA with this supplementation strategy (whole oilseed) impacts on a higher AI, which is related to the higher content of SFA and therefore a less positive human-health effect [86,101]. Roasted oilseeds (whole and ground) are better utilized by ruminants [102], but in our review only had a positive effect on omega-6 PUFA, LA and OA, like the results reported by Rafiee-Yarandi et al. [103], which may be related to changes in structural components that increase the level of protection of FA to ruminal biohydrogenation to some degree [61]. On the other hand, total UFA are affected, especially omega-3 PUFA, which may be due to the increased instability of these FA with the heat of cooking. This effect on UFA (omega-3) resulted in a higher AI, which is undesirable in cow’s milk; however, the type of oilseed to be roasted should be considered; for example, including roasted ground soybeans in the diet of cows resulted in a higher UFA content and lower SFA content, having a lower AI, therefore producing healthier milk [101].

Milk’s fat content and FA composition are strongly affected by the fiber content and F:C ratio. The increasing of the F:C ratio is a strategy to increase the levels of PUFA in milk [104], as is revealed in the meta-analysis carried out by Angeles-Hernandez et al. [105], who reported higher levels of milk fat (>0.32 g 100 g^−1^) and CLA content (>2.28 g 100 g^−1^ FA) in diets with an inclusion of at least 40% DM of forage. Additionally, the consumption of appropriate of levels of high-quality forage to allow the rumen functions to be maintained under optimal conditions [64,106]. From another perspective, diets with a low F:C ratio, and associated with a ruminal pH below 6.0, reduce PUFA biohydrogenation, resulting in alternative routes and changes in the production of biohydrogenation intermediates [104]. In relation to the intervention factor assessed in the current work (oilseed supplementation), an interaction between the F:C ratio and the response to the supplementation of oil sources was reported by Palmquist and Jenkins [60]. Additionally, Ueda et al. [95] observed a significant interaction between a high forage or high concentrate ratio in the diet and flaxseed oil supplementation on ruminal digestion.

The above-mentioned studies support the role of the forage-inclusion level covariate as the source of between-study variability. The significate interaction of this covariate was attributed to the CLA, Σ omega-6, and SFA outcomes. This significant relationship can be explained by the fact that the PUFA-rich oil supplementation decreased de novo synthesis in the mammary gland, which can be associated with the reduction in the synthesis of acetate and butyrate, or with the changing in the hydrogenation pathway generating FA with the subsequent inhibition of lipogenic enzymes [95]. Certainly, Castro et al. [107] assumed a reduced lipogenic activity in the mammary gland due to the effect of the addition of PUFA-rich sources in the diet with mixtures of conserved forages, which is reflected in the lower total fat content in milk compared to the control; in addition, the CLA content in milk was lower with the supplementation of LA compared to LNA, due to the inclusion of soybean and flaxseed oil in the diet, respectively. With respect to CLA synthesis in the mammary gland, the activity of the enzyme delta-9 desaturase is highly correlated with VA content [25]. However, changes in rations, by manipulating the F:C ratio or oil intake, cause modifications in the microbial population, which could alter the ruminal biohydrogenation of MUFA and PUFA, promoting the synthesis of specific isomers, which alters the availability of VA by the mammary gland [64].

In accordance with our findings, the level of NDF of diets affects the response to oil supplementation to fat content, OA and UFA outcomes. In this sense, Sterk [68] stated that the NDF content of the diet affects the UFA in milk content, but the degree of affectation is dependent on the form of supplementation of rich sources of FA in the diet. The effect of dietary NDF content on the UFA content in milk is more negative when the source of UFA in the diet is in free oil form, in contrast to the protected form [68]. A low dietary fiber content is related to less complete biohydrogenation [98], which would explain the higher proportion of UFA in milk fat. Additionally, the same authors [68] indicated that the effect of NDF content depends on the type of main forage in the diet and the UFA content of the diet. The milk FA profile is a product of the manipulation and interaction of a set of factors such as: diet composition, feed intake, ruminal fermentation pattern, lipid metabolism in the liver, body fat mobilization, ruminal biohydrogenation and bacterial degradation of FA and synthesis and absorption of FA in rumen and mammary glands [92,93]. We therefore suggest that these factors (almost the same as those reported in the current study) must be considered at the farm and industry levels in the design of feed strategies and in the research process to reduce the noise effect, taking into account the role of these factors as covariates when it is possible.

The cross-over design determined the allocation of two or more treatments to the same experimental unit but in different periods. In this sense, there should be no carry-over or lasting effect for the previous treatment. Hence, the purpose of wash-out periods is to eliminate the effect of the previous treatment [107]. Our result revealed significant effects of the experimental design and length of wash-out periods on some of the milk components, mainly those associated with the FA profile. Our meta-regression results revealed that the response to oilseed supplementation decreases as the number of wash-out days increases to LN, OA VA, CLA and UFA outcomes, which could elucidate a possible cumulative effect of previous treatment in the cross-over design when the wash-out period is short. Therefore, to define the duration of wash-out periods, the variable response and nature of the treatment must be taken into account [108]. According to our results, cross-over experiments designed to evaluate the effect of oilseed inclusion on milk composition must consider a minimum wash-out period of 20 days to avoid a type I error.

## 5. Conclusions

In this review, based on our meta-analysis, we found that the higher level of oilseed supplementation had a positive linear effect on the unsaturated fatty acid content, when linseed, rapeseed, and soybean were used. The level of forage in the diet was found to have a strong relationship with the contents of saturated fatty acids in milk. Interestingly, experimental washout periods affect the contents of unsaturated fatty acids, specifically rumenic (CLA), vaccenic, linolenic, and oleic fatty acids. Additionally, our analysis showed that the crossover statistical design affects the content of vaccenic fatty acid in milk fat, while the longitudinal design benefits CLA; however, the latter design positively affects saturated fatty acid contents. Overall, our study could help to refine further research efforts aiming to increase the presence of bioactive fatty acids in milk. The findings from our study could be used to find efficient nutritional strategies in different breeds of cows at different stages of lactation to improve not only milk fatty acids, but also other milk components that are relevant for milk processing.

## Figures and Tables

**Figure 1 animals-12-01642-f001:**
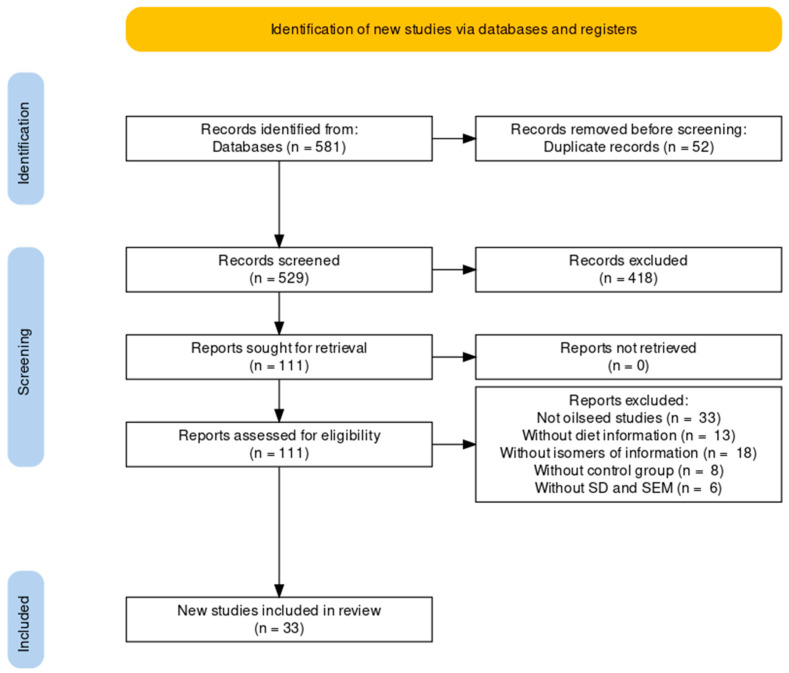
Flowchart of the systematic review (PRISMA) from the initial search to the selection of the articles included in this review.

**Figure 2 animals-12-01642-f002:**
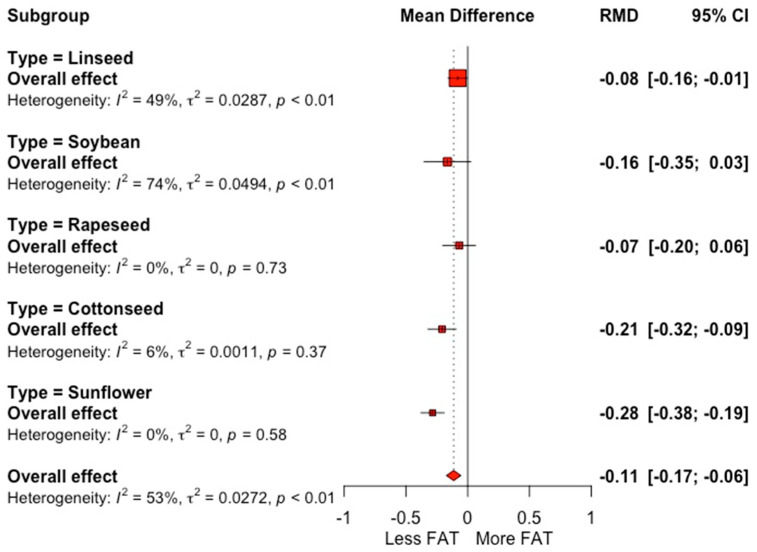
Forest plot of fat content (g 100 g^−1^) for trials with the inclusion of oilseeds in dairy cow diets, grouped by type of oilseed. Point size reflects the relative weighting of the study to the overall effect size estimated, where a larger point size represents a greater weight and the combined effect size estimated, including the confidence intervals. The diamond represents the overall effect. RMD, raw mean difference.

**Figure 3 animals-12-01642-f003:**
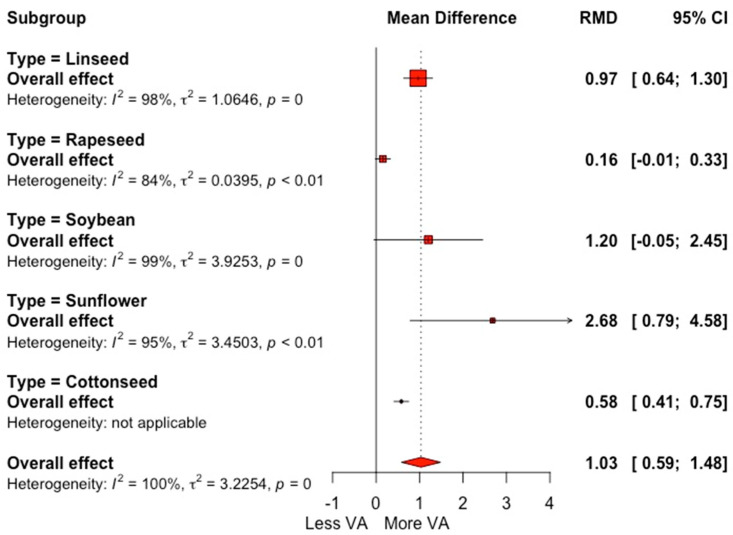
Forest plot of vaccenic fatty acid (VA; g 100 g^−1^ FA) for trials with the inclusion of oilseeds in dairy cow diets, grouped by type of oilseed. Point size reflects the relative weighting of the study to the overall effect size estimated, where a larger point size represents a greater weight and the combined effect size estimated, including the confidence intervals. The diamond represents the overall effect. RMD, raw mean difference.

**Figure 4 animals-12-01642-f004:**
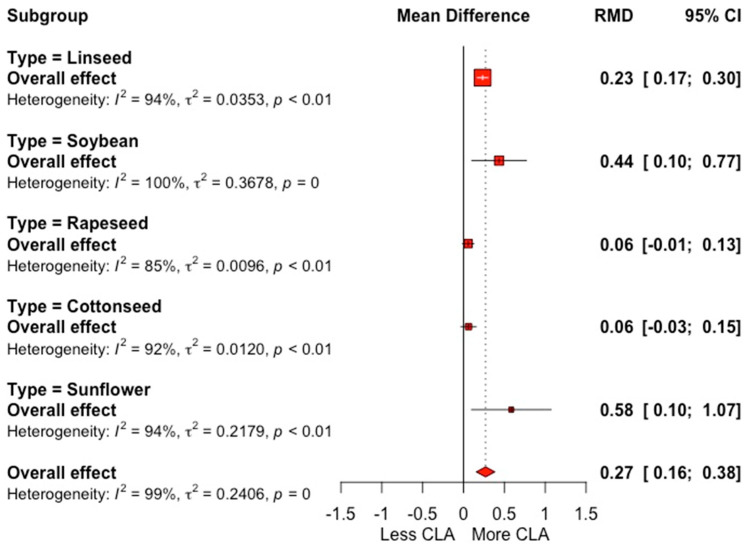
Forest plot of conjugated linoleic acid (CLA; g 100 g^−1^ FA) for trials with the inclusion of oilseeds in dairy cow diets, grouped by type of oilseed. Point size reflects the relative weighting of the study to the overall effect size estimated, where a larger point size represents a greater weight and the combined effect size estimated, including the confidence interval. The diamond represents the overall effect. RMD, raw mean difference.

**Figure 5 animals-12-01642-f005:**
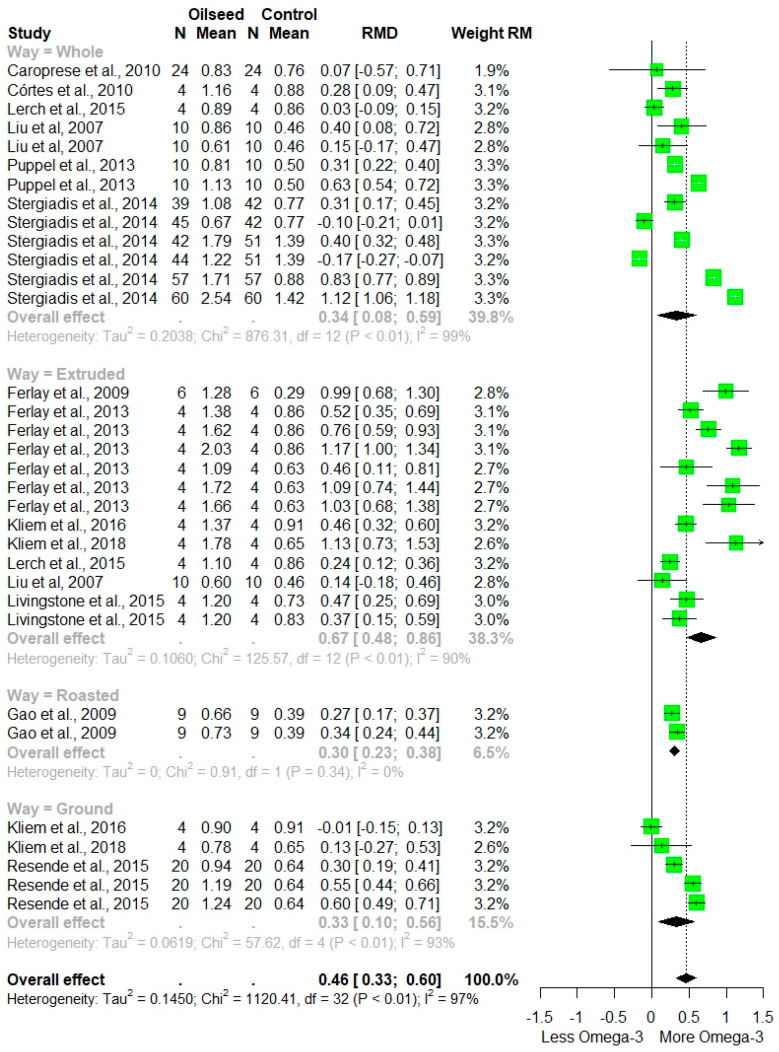
Forest plot of Σ omega-3 PUFA (g 100 g^−1^ FA) for trials with the inclusion of oilseeds in dairy cow diets, grouped by oilseed processing. Point size reflects the relative weighting of the study to the overall effect size estimated, where a larger point size represents a greater weight and the combined effect size estimated, including the confidence intervals. The diamond represents the overall effect. RMD, raw mean difference [25,29,33,35,37,38,40,41,43,48,50,53].

**Figure 6 animals-12-01642-f006:**
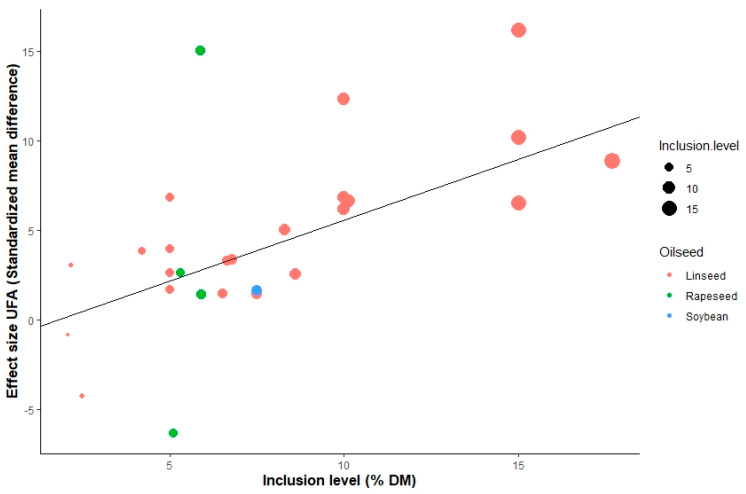
Bubble plot of UFA (g 100 g^−1^ FA) of the raw mean difference. Inclusion level of oilseeds is represented by the size of the point with the value to the left (%); the color of the point indicates the type of oilseed.

**Figure 7 animals-12-01642-f007:**
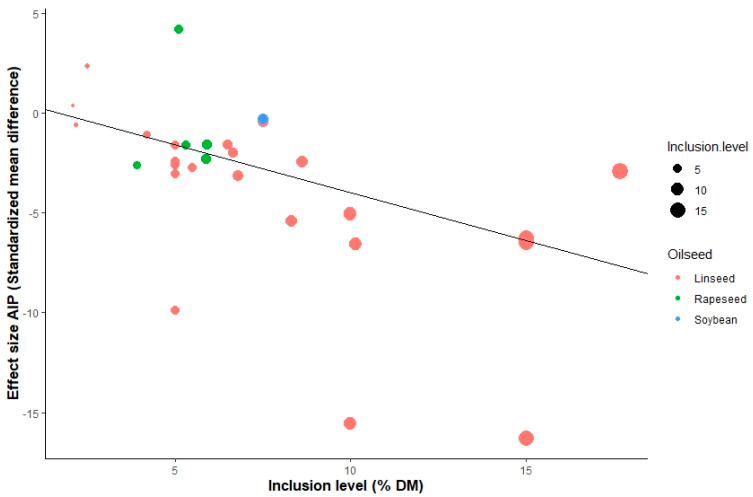
Bubble plot of AI of the raw mean difference. Inclusion level of oilseeds is represented by the size of the point with the value to the left (%); the color of the point indicates the type of oilseed.

**Table 1 animals-12-01642-t001:** Studies, characteristics of animals and oilseed, inclusion, washout and experimental design included in meta-analysis.

Study	Breed ^1^	Lactation Stage	Type ^2^	Way ^3^	Inclusion (%)	Washout (Days)	Design ^4^
[24]	H	2nd	Lin	Wh, Ex	16.4, 16.7	19	Cross
[25]	H	1st	Lin	Wh	6.5	14	Long
[26]	H	1st	Sb, Rap, Cott	Wh, Ex, Gr,	5, 10.1, 10.5	14, 18	Cross, Long
[27]	H	1st	Rap	Gr	14	14	Long
[28]	H	3rd	Lin	Wh	12.4	23	Cross
[29]	H	1st	Lin	Wh	4.2	21	Cross
[30]	H	2nd	Sb, Cott	Ex	11.9	21	Long
[31]	H	2nd	Rap, Lin	Ex, Gr	5, 6.1	21	Cross
[32]	H	1st	Lin	Ex	17.7	21	Long
[33]	H	1st, 2nd	Lin	Ex	5, 10, 15	21	Cross
[34]	H	1st	Lin, Sb	Wh	8, 16	84	Long
[35]	H	1st	Lin, Sb	Rs	7.5	10	Cross
[36]	H	3rd	Lin	Wh, Ex, Gr	12.1, 12.7	21	Cross
[37]	H	3rd	Lin, Rap	Ex, Gr	3.9, 5.5	21	Cross
[38]	H	2nd	Lin, Rap	Ex, Gr	5.9, 8.6	21	Cross
[39]	H	2nd	Sb	Ex	7.5, 5.0	7	Long
[40]	H	1st	Lin, Rap	Wh, Ex	2.5, 5.1	120	Long
[41]	H	2nd	Sb	Wh, Ex	8.5	21	Long
[42]	H	1st	Sb, Lin, Sun, Cott	Rs	7.5	14	Long
[43]	H	1st	Lin	Ex	5	-	Cross
[44]	H	1st	Lin	Wh, Gr	7	14	Cross
[45]	H	1st	Lin	Ex	6.5	14	Cross
[46]	H	1st	Lin	Ex	10	14	Cross
[47]	H	2nd	Lin	Ex, Gr	4.3, 8.1, 8.3, 11.8	14	Cross
[48]	H	2nd	Lin	Wh	2.1, 2.2	7	Long
[49]	H	1st	Sun	Gr	8	14	Cross
[50]	JR	2nd	Lin	Gr	5, 10, 15	14	Cross
[51]	H	3rd	Cott	Wh	12	-	Long
[52]	H	1st	Sun	Wh, Rs	7.8	14	Cross
[53]	H	-	Lin, Rap	Wh	6.6, 6.8, 8.3, 10.5	-	Long
[54]	H	-	Lin, Rap	Wh	8.3	16	Cross
[55]	H, BS	1st	Sb	Ex	10.6	14	Cross
[56]	H	2nd	Sb	Ex	10.7	7	Long

^1^ H = Holstein; JR = Jersey; BS = Brown Swiss; ^2^ Lin = linseed; Rap = rapeseed; Cott = cottonseed; Sun = sunflower; Sb = soybean; ^3^ Wh = whole; Ex = extruded; Gr = ground; Rs = roasted; ^4^ Cross = crossover; Long = longitudinal.

**Table 2 animals-12-01642-t002:** Effect of oilseed inclusion on milk yield, chemical composition and fatty acid profile of dairy cows.

Item	n ^1^	Control Means (SD)	Effect Size				Heterogeneity	
			RMD ^2^	*p*-Value	SMD ^3^	*p*-Value	*I*^2^ RM ^4^	*I*^2^ MM ^5^
Milk yield (kg day^−1^)	73	28.02 (6.59)	−0.25	0.200	−0.06	0.330	50.2	0.0
Fat content (g 100 g^−1^)	70	3.86 (0.46)	−0.11	0.001	−0.21	0.002	53.0	25.8
Protein content (g 100 g^−1^)	69	3.18 (0.2)	−0.04	0.007	−0.20	0.003	65.4	33.1
Lactose content (g 100 g^−1^)	53	4.77 (0.19)	0.01	0.280	0.07	0.270	43.5	19.0
Oleic acid (g 100 g^−1^ FA)	69	20.87 (5.04)	3.44	<0.001	1.50	<0.001	92.4	82.3
Linoleic acid (g 100 g^−1^ FA)	71	2.12 (0.51)	0.09	0.007	0.16	0.250	86.6	83.3
Linolenic acid (g 100 g^−1^ FA)	70	0.49 (0.25)	0.28	<0.001	1.68	<0.001	97.8	76.7
Vaccenic acid (g 100 g^−1^ FA)	70	1.61 (0.88)	1.03	<0.001	1.33	<0.001	99.7	95.2
CLA (g 100 g^−1^ FA)	77	0.54 (0.24)	0.27	<0.001	1.28	<0.001	99.4	78.5
Σ Omega−3 PUFA (g 100 g^−1^ FA)	33	0.74(0.27)	0.46	<0.001	2.23	0.001	97.1	53.0
Σ Omega−6 PUFA (g 100 g^−1^ FA)	33	2.12(0.54)	−0.01	0.780	−0.09	0.530	81.0	48.0
SFA (g 100 g^−1^ FA)	30	63.54(17.03)	−6.51	<0.001	−1.73	<0.001	83.1	7.6
UFA (g 100 g^−1^ FA)	28	28.72(5.95)	8.32	<0.001	3.79	<0.001	93.5	61.7
Atherogenic index (AI)	30	3.01(0.88)	−1.01	<0.001	−2.47	<0.001	91.5	73.9

^1^ Number of trials comparing supplemented and non-supplemented treatments; ^2^ RMD, raw mean difference; ^3^ SMD, standardized mean difference; ^4^ heterogeneity of random effects model; ^5^ heterogeneity of mixed effects model.

**Table 3 animals-12-01642-t003:** Parameters of the meta-regression procedure with the outcome variables and effect size (raw mean difference) between oilseed-supplemented and control treatments of dairy cows.

Item	Meta-Regression Parameters (β)											Best Model	
	Animal		Diet and Oilseeds							Experimental		<*I*^2^	*R* ^2^
	Breed	Stage	Type ^1^	Way ^2^	Intake	Linoleic	Linolenic	Forage	NDF	Washout	Design		
Milk yield (kg day^−1^)				Rs −0.47 *; Wh −0.28 *	−0.05 ***							100.0	100
Fat content (g 100 g^−1^)	JR 0.38 *	2nd 0.39 *	Sun −0.46 *	Wh 0.21 *			0.01 *		−0.07 *			51.3	57.8
Protein content (g 100 g^−1^)		1st 1.57 **; 3rd 1.57 *				0.02 *						49.3	53.1
Lactose content (g 100 g^−1^)			Lin −0.09 *; Rap −0.10 *			−0.005 ***						56.3	74.7
Linoleic acid (g 100 g^−1^ FA)	JR −0.61 *	1st 0.49 *	Rap −0.96 *	Gr 0.43 *	−0.02 *							3.8	11.8
Linolenic acid (g 100 g^−1^ FA)		1st −2.4 *; 2nd −2.33 * 3rd −3.79 **		Rs 4.16 **	−0.12 *	−0.13 ***				−0.02 *		21.5	33.5
Oleic acid (g 100 g^−1^ FA)		3rd −4.07 *	Rap 6.71 *: Sb 3.54*; Sun 4.92 **	Gr 2.63 *; Rs 4.41 **; Wh 2.61 *	0.33 *	−0.23 ***	−0.15 ***		−0.07 *	−0.05 **		10.9	9.8
Vaccenic acid (g 100 g^−1^ FA)			Lin −3.79 ***; Rap −3.92 ***; Sb −3.11 ***			−0.04 *				−0.02 ***	Cross −1.31 *	4.6	95.2
CLA (g 100 g^−1^ FA)			Lin 1.29 *; Sb 1.65 *; Sun 1.96 *	Wh −0.18 **				−0.09 *		−0.03 **	Long 1.45 ***	21.0	21.1
Σ Omega-3 PUFA (g 100 g^−1^ FA)	JR 1.15 *		Rap 1.35 *	Gr −1.38 *; Rs −0.74 *	0.04 *							45.4	50.7
Σ Omega-6 PUFA (g 100 g^−1^ FA)			Sb 0.67 *	Rs 0.49 **	−0.01 *			−0.11 *				40.7	75.1
SFA (g 100 g^−1^ FA)				Wh 4.25 ***	−0.37 **			3.26 *			Long 4.85 **	90.7	96.7
UFA (g 100 g^−1^ FA)		1st −7.31 **	Rap 12.25 **	Rs −5.43 *; Wh −10.98 *; Gr −14.66 **	0.79 *		−0.08 *		0.19 ***	−0.15 ***		34.0	98.4
Atherogenic index (AI)		1st −0.89 *; 2nd 1.15 *		Rs 1.81 ***; Wh 1.23 ***	−0.12 *					0.01 **		19.2	93.9

* *p* < 0.05, ** *p* < 0.01, *** *p* < 0.001. JR = Jersey; ^1^ Lin = linseed; Rap = rapeseed; Cott = cottonseed; Sun = sunflower; Sb = soybean; ^2^ Wh = whole; Ex = extruded; Gr = ground; Rs = roasted; Intake, level of oilseed inclusion in the diet; Linoleic, difference of linoleic acid content of supplemented and control rations; Linolenic, difference of linolenic acid content of supplemented and control rations; Forage, difference of forage inclusion level of supplemented and control rations; NDF, difference of NDF of supplemented and control rations; Washout, days before the measurement of variables.

## Data Availability

Data available upon request to the corresponding author.

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
