# Peer review of "Oilseed Supplementation Improves Milk Composition and Fatty Acid Profile of Cow Milk: A Meta-Analysis and Meta-Regression"

_animals, 2022, doi:10.3390/ani12131642_

Round 1

Reviewer 1 Report

The manuscript is a review that uses meta-analysis and meta-regression to present the effects of oilseed supplementation in milk production and milk composition, with a special focus on the fatty acid of cow milk that somehow has an impact on human health. The introduction section focuses on CLA but is consistent. The results are clearly described and easy to understand. The discussion section needs some improvement. Same specific considerations are below:

Line 24-25. It is not clear in the text what it has a higher content of 18-carbon unsaturated. Is It the oilseed supplementation? Please, be specific.

Figure 1. Improve the quality of the figure.

Table 2. Milk yield (l/day). Please double-check the unit. The most common one is kg/day. Probably it is wrong in the paper.

Line 376-380. The authors have tried to explain the reduction of milk protein concentration by a decrease of ruminal microorganisms caused by the presence of UFA in the diet. It is well known that this is possible, however, it’s highly dependent on the concentration of FA in the diets and the profile of FA, mostly UFA. The inclusion level of oilseed is down to 150 g/kgDM top, and most of the inclusion levels are down to 100 g/kgDM. In this sense, the authors should use another approach to explain the suggestive reduction in milk protein. What was the variation in the CP of the control and treatment diets?

Line 381. Change “behavior” to parameters.

Line 413-429. May an approach with a possible reduction in propionate in the rumen could be interesting. Once the fat milk content is reduced. Please, include something in this sense.

Line 437-442. The authors should include some comments about the number of papers with each breed. It is very clear that the data are not 100% reliable once there is just one paper with Jersey and one with Brown Swiss.

Reviewer 2 Report

The study reports a metanalysis on the effects of oils supplementation in the diets of dairy cows on fat compositions of milk.

The stastisical approach is not adquately described and requires additional information. The introduction and discussion of the mansucript often focuses the attention of the role of milk fat composition on human health, but the dietary positive effect of the increase of “desirable” fatty acids, or the variation of AI index, in the milk is questionable. Can this variation really impacts on the intakes and justifies the addition of oil in the diet of dairy cows?   

Lines 16-17: Are you sure that mil is the major source?

Line 36: stage?

Line 42: what 0.27 g/100 g FA means (and the others >)?

Line 47: Desirable, why?

Lines 58-59: References are needed to support this sentence.

Line 63: 2 to 53.7 mg g-1 of fat, in milk?

Line 73: “Dietary manipulation of dairy cows” maybe Manipulation of diets fed to dairy cows

Line 78: lower cost, higher UFA in comparison to?

Line 99: In the line 32, web of science is reported, not PubMed.

Line 153: The metaregression analysis include explanatory variables that are continuous (intake, linoleic, linolenic, forage, NDF) and discrete (breed, stage of lactation, type, washout). Add more information on  the model used.  The meta regression analysis considers main effects, but not interaction. How the authors account for that?

Figure 2 and others: Report RMD in the caption.

Figures from 2 to 5 present odd ratios of main effects. In the materials and methods are not reported how were calculated.

Conclusion

More than a conclusion, is a summary or the results. Reword this section underlining the take home message of the use of oil in diets for dairy cows.

Round 2

Reviewer 2 Report

The manuscript was improved and explanations were added. I still have some concern for the milk consumption and human health. As indicated by the authors, the average positive effect of oilseed on CLA was 0.27 g/100 g fat and on UFA +8.32 g/100 g fat. That means that if one consumes 0.15 kg of milk every day (average for EU citizen on 2021) with 4% of fat the expected increase of CLA is from 0.032 to 0.049 g/d and for UFA from  1.72 to  2.22. Can this increase justify a positive role of milk on human health?  
